# Pain Catastrophizing in Older Adults with Chronic Pain: The Mediator Effect of Mood Using a Path Analysis Approach

**DOI:** 10.3390/jcm9072073

**Published:** 2020-07-01

**Authors:** Huan-Ji Dong, Björn Gerdle, Lars Bernfort, Lars-Åke Levin, Elena Dragioti

**Affiliations:** 1Pain and Rehabilitation Centre, and, Department of Health, Medicine and Caring Sciences, Linköping University, SE-581 85 Linköping, Sweden; huanji.dong@liu.se (H.-J.D.); bjorn.gerdle@liu.se (B.G.); 2Division of Health Care Analysis, Department of Health, Medicine and Caring Sciences, Linköping University, SE-581 85 Linköping, Sweden; lars.bernfort@liu.se (L.B.); lars-ake.levin@liu.se (L.-Å.L.)

**Keywords:** pain catastrophizing, anxiety, depression, mediate, older people

## Abstract

Cognitive models of pain propose that catastrophic thinking is negatively associated with chronic pain. However, pain catastrophizing is a complex phenomenon requiring a multivariate examination. This study estimates the effects of mood variables (anxiety and depression) on pain catastrophizing in older adults with chronic pain. A postal survey addressing pain aspects was sent to 6611 people ≥ 65 years old living in south-eastern Sweden. Pain catastrophizing was measured using the pain catastrophizing scale. Anxiety and depression were assessed using two subscales of the general well-being schedule. Data were analysed using a path analysis approach. A total of 2790 respondents (76.2 ± 7.4 years old) reported chronic pain (≥three months). The mediation model accounted for 16.3% of anxiety, 17.1% of depression, and 30.9% of pain catastrophizing variances. Pain intensity, insomnia, number of comorbidities, and lifestyle factors (smoking, alcohol consumption, and weight) significantly affected both pain catastrophizing and mood. Anxiety (standardized path coefficient (b_std_) = 0.324, *p* < 0.001) in comparison to depression (b_std_ = 0.125, *p* < 0.001) had a greater effect on pain catastrophizing. Mood mediated the relationship between pain catastrophizing and pain-related factors accounting for lifestyle and sociodemographic factors.

## 1. Introduction

Although pain is a universal experience throughout the life course, chronic pain in later life is an increasing global health problem considering the rapid growth of older populations. The prevalence of chronic pain is known to increase with age, ranging between 25% and 76% in the general elderly population and up to 93% of elderly in residential care [1,2]. In Sweden, more than 50% of people aged 65 and over report chronic pain (irrespective of intensity) [3].

Chronic pain is generally not a symptom or complaint that exists in isolation. For older people, considerable evidence connects chronic pain with comorbidities, including mood disorders (anxiety and/or depression) [4,5,6,7] and sleep disturbance [8,9]. In this context, one’s psychological state is an important determinant of pain experiences as well as mental health [10,11,12]. Brain images indicate that cognitive and emotional modulations of pain are associated with alterations in specific brain regions [13,14]. Clinically, psychological factors such as anxiety, catastrophizing, and depressive symptoms seem to be important features of patients with chronic pain [11,15]. Pain catastrophizing—a persistently negative cognitive affective style characterized by helplessness, magnification, and ruminative thoughts regarding one’s pain—is a potent predictor of negative pain-related outcomes in general [16]. Vlaeyen and Lintons’ fear-avoidance model, perhaps the most known framework, highlighted pain catastrophizing as a cognitive precursor of fear and avoidance and a possible misinterpretation of pain [17]. The fear-avoidance model illustrated the transition from a common pain episode toward persistent pain and disability via pain catastrophizing [18]. In brief, pain catastrophizing is associated with pain severity, disability, and poor outcomes for patients with chronic pain [14,19,20,21]. Evidence suggests that pain catastrophizing in older adults is associated with sedentary behaviour which in in turn, contributed to greater pain catastrophizing [22]. Catastrophizing significantly predicts the development of chronic pain in pain free individuals as well as the chronification of acute pain [19]. Treatment studies show that initial decreases in pain catastrophizing can predict subsequent changes in pain intensity and/or pain interference [20,21]. In addition, reductions in pain intensity and/or interference early in treatment are associated with subsequent reductions in catastrophizing in patients with neuropathic pain [20]. Together, these results indicate complex interrelationships between pain and catastrophizing. A recent systematic review and meta-analysis concerning pain catastrophizing in chronic pain found weak associations with pain intensity and disability from both cross-sectional and longitudinal perspectives [23]. The meta-analyses indicated prominent heterogeneity. The authors suggest that moderators influence the strength of the associations between pain catastrophizing and pain intensity/disability and warrant research including large cohorts and adjusting for all covariates.

The relationship between pain catastrophizing and emotional processing (i.e., worry, anxiety, and depression) is not clear. Some studies suggest that pain catastrophizing mediates the relationship between pain and depressed mood [24,25,26,27,28]. One study highlights that depressive and anxiety symptoms mediate the relationship between pain catastrophizing and pain [29]. One disadvantage of these studies is that they do not always consider lifestyle and sociodemographic factors, which have been found to have significant effects on pain aspects and mood disorders [30,31,32]. Furthermore, sleep disturbance, especially insomnia, should also be considered due to its complex associations with pain and mood disorders [33,34,35].

In pain rehabilitation, catastrophizing is described as a maladaptive coping strategy [36,37]. Improvement of maladaptive coping strategies is an important goal of comprehensive pain rehabilitation programs. To achieve good rehabilitation results, the comprehensive rehabilitation process (i.e., interdisciplinary pain treatment according toInternational Association for the Study of Pain; IASP) considers many factors, focusing on the whole person rather than biomedical factors [36].

Catastrophizing is more strongly associated with pain intensity among older people, whereas pain intensity among younger people is more strongly associated with emotional responses to pain [38]. In addition, older people with chronic pain show less passive coping and higher life control than younger people with chronic pain [39]. Longitudinal data reveal that the predictive effect of catastrophizing on worse pain and disability disappeared when adjusting for age, gender, and mood disorders [29]. Hence, it is reasonable to expect that older people will exhibit other patterns associated with pain, mood, and pain catastrophizing. To efficiently treat patients who catastrophize chronic pain, it is necessary to discover the factors, particularly modifiable factors, that affect catastrophizing. Based on the fear avoidance model cognitive-behavioural risk factors play a crucial role in the development and persistence of pain. However, there is ongoing evidence supporting the necessity for modifications on the model to address how multi-dimensional factors may interact [40,41]. Therefore, an examination of all causal relationships is needed to better understand the key factors that influence pain catastrophizing in older adults.

This study estimates the effects of mood variables (anxiety and depression) on pain catastrophizing in older adults with chronic pain. We also explore the impact of pain intensity and insomnia on mood variables controlled for lifestyle and sociodemographic factors. We hypothesize that these relationships are mediated by mood variables (i.e., anxiety and depression).

## 2. Materials and Methods

### 2.1. Study Population

Cohort PainS65+, a Swedish population-based study, focuses on pain aspects and health experiences in the elderly. Data collection for Cohort PainS65+ was a cross-sectional design based on the Swedish Total Population Register (STPR) for the two large cities (Linköping and Norrköping) in a south-eastern county (Östergötland) of Sweden [11,30,42,43]. The STPR consists of ~49,320 older adults and a stratified random sample of 10,000 older adults (≥65 years old) in five age strata (65–69 years, 70–74 years, 75–79 years, 80–84 years, and 85 years and older) was selected by Statistics Sweden (SCB). A postal survey was conducted between October 2012 and January 2013. Two reminders at two-week intervals were mailed if necessary. The study was approved by the Regional Ethics Research Committee in Linköping, Sweden (Dnr: 2012/154-31).

### 2.2. Measurements

An overview of the validated instruments/scales of the survey has been presented elsewhere [11,30,42,43]. The relevant instruments for this study are described below.

#### 2.2.1. Sociodemographic Data

Age (years), sex (men/women), marital status (married /not married), and highest educational level (university/not) were recorded from the respondents’ answers in the postal survey.

#### 2.2.2. Chronic Pain

We began by asking the following question: Do you usually have pain, either all the time or occasionally? Three answer alternatives were provided: no; yes, with less duration than three months; and yes, with a duration of more than three months. Respondents who selected the third alternative answer were identified as participants with chronic pain. This study presents all other characteristics for this sample.

#### 2.2.3. Pain Intensity

Participants were asked to rate their pain intensity over the previous seven days using an 11-point numeric rating scale (NRS-7d), ranging from 0 (no pain) to 10 (worst imaginable pain) [44].

#### 2.2.4. Pain Catastrophizing

The pain catastrophizing scale (PCS) was used to quantify the catastrophic thinking related to pain [45]. Participants were asked to assess the degree to which they experience certain thoughts or feelings during pain. The PCS consists of 13 items and each item has five answer alternatives on a five-point scale, ranging from 0 (not at all) to 4 (always). A total score (0–52) was determined along with three subscale scores assessing rumination, magnification, and helplessness. We used the total score. However, due to a printing issue, the most negative alternative (4: all the time) was not included in the questionnaire. Therefore, the PCS resulted in a total possible score of 39. We also estimated the reliability of the instrument by calculating the Cronbach alpha (a), and reliability was good (a = 0.75) [11].

#### 2.2.5. Mood (Anxiety and Depression)

We selected the items from the general well-being schedule (GWBS) to assess the mood state of the participants. As developed by Fazio, the GWBS measures psychological well-being and distress over a month [46]. The instrument provided good internal consistency, test-retest reliability, and validity [47,48]. The GWBS can be divided into six subscales. In this study, two subscales, namely anxiety (4 items; range: 0–25) and depression (3 items; range: 0–20), were used and treated as continuous variables. For the subscales of anxiety and depression, low values indicate higher anxiety and depression. These items were reverse scored, so high values indicate higher symptoms of anxiety and depression.

#### 2.2.6. Insomnia

The insomnia severity index (ISI) was used to assess sleep problems. ISI is a reliable and valid instrument for detecting cases of insomnia and has excellent internal consistency [49,50]. Each item is rated on a five-point Likert scale (0–4). A sum of the seven items generates a score between 0 and 28.

#### 2.2.7. Lifestyle Factors

Smoking: We used the instrument Health Curve (Hälsokurvan) [51] to gather data on health behaviours. Participants were asked about smoking habits including frequency (from never to daily). The variable is denoted as current smokers, ex-smokers, or never smokers.

#### 2.2.8. Alcohol Consumption

Participants were asked about their alcohol consumption: Do you drink alcohol regularly? The response was yes or no. For those who replied with yes, we added the four CAGE (Cut-down, Annoy, Guilty, and Eye-opener) questions to evaluate the possible addiction problems [52]. A score between 0 and 1 was defined as low consumption and a score ≥ 2 was regarded as high consumption, indicating potential problems with alcohol addiction.

#### 2.2.9. Weight Status

Body mass index (BMI = weight (kg)/height (m)^2^) was calculated based on self-reported body height and weight. BMI was classified according to the criteria developed by the World Health Organization (WHO): <18.5 = underweight; 18.5–24.9 = normal range; 25.0–29.9 = overweight; 30.0–34.9 = obesity I; and ≥35.0 = obesity II and III (severe obesity).

#### 2.2.10. Comorbidity

The presence of comorbidity was also captured in the survey. We used a self-reported questionnaire (yes or no responses) covering 12 physical and psychological disorders/conditions over the previous two weeks: traumatic injuries, rheumatoid arthritis and osteoarthritis, cardiovascular disorders, pulmonary disorders, depressive disorders, anxiety disorders, gastrointestinal disorders, central nervous system disorders, urogenital disorders, skin disorders, tumours and cancers, and metabolic disorders [30]. The sum of all confirmed comorbidities for each participant was denoted as number of comorbidities, ranging from 0–12.

### 2.3. Statistical Analysis

The statistics were performed using the statistical package IBM SPSS Statistics (version 25.0; IBM Inc., New York, USA) and R statistical language and environment (version 3.6.1) using the lavaan package [53]. R is available as free software under the terms of the Free Software Foundation’s GNU General Public License in source code form. All tests with *p* < 0.05 were considered statistically significant. Descriptive statistics were calculated for all variables. Pearson’s r was used to examine the bivariate relationships among the psychometric scales. Analysis of covariance (ANCOVA) was used to examine the influence of lifestyle and sociodemographic factors on anxiety, depression, and pain catastrophizing. In the next step, the subscales anxiety and depression were examined for their influence on the pain catastrophizing using a path analysis model. Path analysis is a form of multiple regression analysis used to evaluate causal models by examining the relationships between a dependent variable and two or more independent variables. This method estimates both the magnitude and significance of causal links between variables. We used standardized path coefficients (b_std_) to estimate effect size [54]: absolute values less than 0.10 indicate a small effect, values around 0.30 indicate a medium effect, and values greater than 0.50 indicate a large effect.

In the path model, the independent variables were pain intensity, insomnia, and lifestyle and sociodemographic characteristics, the dependent variable was pain catastrophizing, and the mediators were anxiety and depression. The basic rules as described in the literature were considered when defining the theoretical model [55,56,57]. To find possible multivariate outliers, we computed the mahalanobis distance (MD) for all cases. The MD measures distance relative to the centroid—a base or central point that can be thought of as an overall mean for multivariate data. The larger the MD, the further away the data point is from the centroid [58].

We tested the path model using the maximum likelihood estimation using the fit indices proposed by Hu and Bentler [59] as well as Barrett [60]. More analytically, we used the Chi-Square (χ^2^) value, which is the traditional measure for evaluating overall model fit and ‘assesses the magnitude of discrepancy between the sample and fitted covariances matrices [59]. A good model fit should provide an insignificant result at a 0.05 threshold [60]. Other indicators were the Tucker Lewis index (TLI), the normed fit index (NFI), the non-normed fit index (NNFI), the comparative fit index (CFI), the goodness-of-fit index (GFI), and the adjusted goodness-of-fit index (AGFI), which shows the model fit relative to the null model. Typically, all indices are considered acceptable when estimates ≥ 0.90 [59]. We also included the root mean square error of approximation (RMSEA) and the standardized root mean square residual (SRMSR). For both indices, estimates ≤ 0.05 were considered a good fit.

## 3. Results

The postal survey was completed and returned by 6,611 older adults, corresponding to a response rate of 66.1%, of which 42% reported chronic pain. Data on the survey characteristics for both the whole sample population and respondents without chronic pain have been described in detail elsewhere [11,30,42]. Therefore, our study sample consisted of 2,790 older adults (61.1% women) with chronic pain with partial missingness for 117 (4.2%), 89 (3.2%), 45 (1.6%), and 32 (1.1%) for smoking, education, alcohol, and BMI, respectively. The mean age was 76.2 years (SD: 7.4), 55.8% were married (Table 1), 8.1% smoked, and 5.9% had high alcohol consumption. Details of the demographic characteristics in this study sample (*n* = 2,790) are also described elsewhere [61,62]. Approximately 40% were overweight and over 20% were classified as obese. The median number of comorbidities was 2.0 (interquartile range (IQR): 1.9–3), ranging from 0 to 11 (Table 1). All the psychometric scales were significantly positively correlated (Appendix A). Τhe analysis of covariance (ANCOVA) showed that all independent variables were associated with at least one dependent or moderator variable, so all variables were selected for the path analysis (Table 2).

The MD calculation found 205 outliers (Appendix A), which were excluded. Hence, the path analysis included 2229 individuals with chronic pain. We formulated and tested the theoretical model as presented in Figure 1. The model offers the possibility of examining all possible interrelations among dependent variable, independent variables, and moderator variables (i.e., anxiety and depression).

The model had a very good fit with a non-significant Chi-square (χ^2^ (4) = 6.212, *p* = 0.184), indicating that the assumed path model is adequate for the data (i.e., the model and the data are not statistically significantly different). The output of the model indicated a very good fit to the data: χ^2^ (45) = 3490.184 (*p* < 0.001); Tucker Lewis index (TLI) = 0.992; normed fit index (NFI) = 0.998; non-normed fit index (NNFI) = 0.992; comparative fit index (CFI) = 0.999; goodness-of-fit index (GFI) = 0.998; adjusted goodness-of-fit index (AGFI) = 0.931; standardized root mean square residual (SRMR) = 0.005; and root mean square error of approximation (RMSEA) = 0.016 (95% C.I. 0.000–0.038). The model accounted for 16.3% of anxiety, 17.1% of depression, and 30.9% of pain catastrophizing variances (Table 3).

### 3.1. Direct Effects on Pain Catastrophizing

The overall findings from path analysis are illustrated in Table 4, and the standardized regression coefficients (b_std_) are presented in Table 3. None of the sociodemographic factors (age, gender, marital status, and education) had a direct significant effect on pain catastrophizing. The lifestyle factors current smokers, ex-smokers, and obesity II and III had a significant direct positive effect on pain catastrophizing (b_std_ = 0.040, *p* = 0.048; b_std_ = 0.041, *p* = 0.029; and b_std_ = 0.051, *p* = 0.012, respectively), and low alcohol consumption (compared to non-drinkers) had a significant direct negative effect on pain catastrophizing (b_std_ = −0.077, *p* < 0.001). Number of comorbidities, pain intensity, and insomnia also had significant positive effects on pain catastrophizing (b_std_ = 0.049, *p* = 0.014; b_std_ = 0.234, *p* = 0.000; and b_std_ = 0.052, *p* = 0.012, respectively). Both anxiety and depression had a significant direct positive effect on pain catastrophizing (b_std_ = 0.324, *p* < 0.001; b_std_ = 0.125, *p* < 0.001), although anxiety had a greater effect than depression.

### 3.2. Indirect Effects on Pain Catastrophizing

These results are presented in Appendix A. Female gender and number of comorbidities had a significant indirect positive effect on pain catastrophizing through pain intensity (*p* = 0.032 and *p* < 0.001). Low alcohol consumption had a significant indirect negative effect on pain catastrophizing through pain intensity (*p* = 0.003).

### 3.3. Direct Effects on Anxiety

These results are presented in Table 3. Age, being a woman, number of comorbidities, pain intensity, and insomnia had a direct significant positive effect on anxiety (b_std_ = 0.073, *p* =0.01; b_std_ = 0.062, *p* = 0.002; b_std_ = 0.122, *p* < 0.001; b_std_ = 0.165, *p* < 0.001; and b_std_ = 0.23, *p* < 0.001, respectively), while overweight and low alcohol consumption (compared to non-drinkers) had a direct negative effect on anxiety (b_std_ = −0.047, *p* = 0.022 and b_std_ = −0.079, *p* < 0.001, respectively).

### 3.4. Direct Effects on Depression

These results are presented in Table 3. Age, being a woman, current smoking, number of comorbidities, pain intensity, and insomnia had a direct significant positive effect on depression (b_std_ = 0.169, *p* < 0.001; b_std_ = 0.040, *p* = 0.048; b_std_ = 0.063, *p* = 0.004; b_std_ = 0.108, *p* < 0.001; b_std_ = 0.092, *p* < 0.001; and b_std_ = 0.216, *p* < 0.001, respectively). University education, overweight, and low alcohol consumption had a direct significant negative effect on depression (b_std_ = −0.054, *p* = 0.007; b_std_ = −0.048, *p* = 0.021; and b_std_ = −0.101, *p* < 0.001, respectively).

## 4. Discussion

The present study addresses the pain catastrophizing and mood (anxiety and depression) in older adults with chronic pain affected by pain intensity, insomnia, comorbidities, as well as lifestyle and sociodemographic factors. Using a mediation analysis to control for lifestyle and sociodemographic factors, we found direct and/or indirect effects on both pain catastrophizing and mood. Mood aspects mediate the relationships between pain catastrophizing and other factors. Anxiety had the largest effect on pain catastrophizing.

Pain catastrophizing was directly affected by pain intensity, insomnia, and number of comorbidities. In line with other studies, these variables also had direct effects on mood [63,64,65,66]. The significant contribution of pain intensity on pain catastrophizing is not only its direct influence but also its availability for other variables to affect pain catastrophizing indirectly, despite their relatively weaker effects in comparison with the direct contributions. Through pain intensity, the number of comorbidities also showed indirect effects on pain catastrophizing. This finding suggests some painful comorbidities influence pain catastrophizing. Additionally, as multimorbidity is more common in older people than in younger people, pain catastrophizing may be more preferentially related to pain intensity.

Lifestyle and sociodemographic factors (female gender and low alcohol consumption) via pain intensity also had indirect impacts on pain catastrophizing. It should be noted, sex difference or alcohol habits affect pain catastrophizing as well as pain intensity. Surprisingly, insomnia did not affect pain catastrophizing through pain intensity. In younger patients with chronic pain, insomnia had, via pain, indirect effects on pain catastrophizing [35]. Low correlation between insomnia and pain intensity has also been reported in the clinical patient cohorts [34]. As older people’s sleep problems are usually multifactorial rather than pain specific [67], sleep problems in older patients deserve more attention, regardless of whether pain intensity is affecting pain catastrophizing. That is, older patients need treatment strategies that younger patients may not.

Our model shows that mood aspects have strong direct effects on pain catastrophizing. A preclinical study suggested that mood had negative effects on cognition and pain [68]. Particularly, we found that anxiety had a larger effect than depression, which is partly reflected by the conceptual work of Flink and his colleges that pain catastrophizing is similar to worries and therefore overlaps with anxiety [69]. Notably, anxiety and depression had high correlation to each other, indicating non-trivial symptoms overlapping in older adults with chronic pain. Therefore, it is reasonable to expect that similar factors influenced both anxiety and depression.

Chronic pain often coexists with anxiety, depression, and pain catastrophizing [38,70,71]. Their concurrent associations as well as the predictive value of pain catastrophizing on poor outcomes for patients with chronic pain have been demonstrated in previous studies [14,19,20,21,23]. Changes in levels of pain catastrophizing by reducing emotional distress and maladaptive behaviours can result in better pain-related outcomes [72]. In this study, we also noted that several factors—including pain intensity, insomnia, comorbidities, and low alcohol consumption—directly affected pain catastrophizing and mood variables. Moreover, some lifestyle and sociodemographic factors significantly affected mood in different directions. To explore how mood mediates pain catastrophizing, our model controlled the parallel influences such as the effects of pain intensity and insomnia on mood and the effects of lifestyle and sociodemographic factors on mood as well as pain catastrophizing. We also quantified the magnitude and significance of links between variables. Clearly, the complex relationship between pain aspects, mood, and pain catastrophizing were influenced simultaneously by lifestyle and sociodemographic factors. Understanding these relationships provides us with the possibility of establishing prevention and management strategies for chronic pain in old age. These relationships include the following: never being a smoker associated with less pain catastrophizing in later life, smoking cessation is associated with decreased depressive symptoms, and weight control is associated with less pain catastrophizing and better mood. Interestingly, little but not too much alcohol (low alcohol consumption) is associated with better mood and less pain catastrophizing in this older population. The debate about pros and cons of alcohol consumption is long lasting [73,74,75]. A low dose of alcohol in contrast to a higher dose seems to be beneficial with respect to mood and pain catastrophizing, but it also might indicate a harmful effort to cope with emotional pain [76].

This study has several limitations. First, a cross-sectional study design does not allow for forming conclusions about causality. We could not know whether anxiety and depression developed after chronic pain or the mood disorder existed before pain debut. We did not know whether older people with chronic pain used alcohol as a strategy to cope with pain, pain catastrophizing or mood disorder. Thus, we assumed a linear association among the variables included in the path analysis model. Second, the non-response bias should be considered as frail older people often have severe impaired cognitive function. The study procedure might also have biased the voluntary sample, because the study design unintentionally selected participants who were strongly interested in the pain and pain-related issues. Therefore, the generalization of our results has its limitations. Third, although we measured most factors that negatively influence chronic pain, we did not measure some important psychological aspects, especially positive or protective factors (i.e., self-efficacy, locus of control, and cognitive coping and appraisal) [27,77]. Finally, the technical error in the response alternatives of the pain catastrophizing scale accompanied by the medium response rate may underestimate our results. In addition, different pain conditions could have distinct psychological profiles [37,78] so future research ought to consider these aspects too.

## 5. Conclusions

This study highlights the importance of pain catastrophizing in older adults with chronic pain. Mood aspects play mediating roles in the relationship between pain catastrophizing and other pain-related factors (pain intensity, insomnia, and comorbidities), accounting for lifestyle and sociodemographic factors. Our findings also stress the importance of treating emotional distress in older adults with pain to increase awareness of potentially harmful self-coping strategies (e.g., alcohol) in this target group. Overall, these findings provide for the possibility of pain prevention and management strategies targeted for older patients.

## Figures and Tables

**Figure 1 jcm-09-02073-f001:**
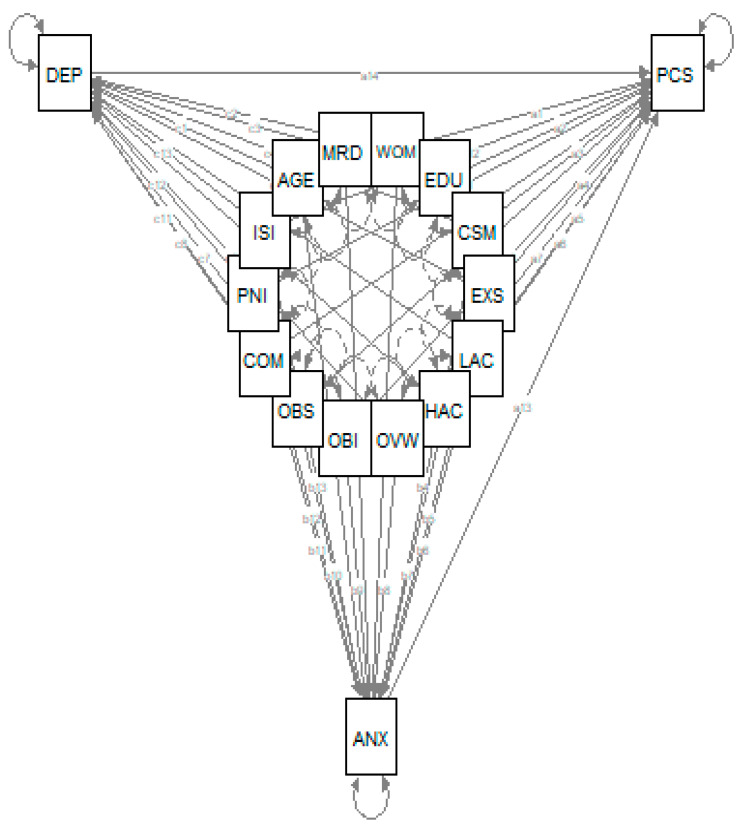
Theoretical model. PCS: Pain Catastrophizing Scale; ANX: anxiety as measured by General Well-being Schedule; DEP: depression as measured by General Well-being Schedule; PNI: Pain Intensity as measured by numeric rating scale for the previous seven days; ISI: Insomnia Severity Index; WOM: Women; MRD: Married; EDU: University education; CSM: Current smokers; EXS: Ex-smokers; OVW: Overweight; OBI: Obesity I; OBS: Obesity II and III; LAC: Low alcohol consumption; HAC: High alcohol consumption; COM: Number of comorbidities.

**Table 1 jcm-09-02073-t001:** Descriptive characteristics of the sample.

Characteristic	*N* (%)
Age, mean (SD)	76.2 (7.4)
**Sex**	
Men	1085 (38.9)
Women	1705 (61.1)
**Married**	
Yes	1557 (55.8)
No	1233 (44.2)
**University education**	
Yes	1213 (44.9)
No	1488 (55.1)
**Smoking**	
Current smokers	226 (8.5)
Ex-smokers	1041(38.9)
Never smokers	1406 (52.6.)
**BMI**	
Low/Normal weight (<25)	1134 (43.1)
Overweight	1017(38.6)
Obesity I	387 (17.4)
Obesity II and III (≥35)	95 (3.6)
**Alcohol consumption**	
Low	1703 (62.0)
High	161 (5.9)
No	881 (32.1)
**Number of comorbidities, median (IQR)**	2.0 (1.9-3.0)
ANX, mean (SD)	6.2 (4.6)
DEP, mean (SD)	5.5 (3.6)
PNI, mean (SD)	4.9 (2.0)
ISI mean, (SD)	10.2 (3.5)
PCS, mean (SD)	13.3 (7.4)

PCS: Pain Catastrophizing Scale; PNI: Pain Intensity as measured by numeric rating scale for the previous seven days; ANX: anxiety as measured by General Well-being Schedule; DEP: depression as measured by General Well-being Schedule; ISI: Insomnia Severity Index; BMI: Body Mass Index; IQR: Interquartile range SD: Standard Deviation.

**Table 2 jcm-09-02073-t002:** Effect of sociodemographic factors on anxiety, depression subscales, and pain catastrophizing scale ^§^.

Factor/Scale	ANX	DEP	PCS
Age (AGE)	<0.001	<0.001	ns
Women (WOM)	<0.001	0.027	0.005
Married (MRD)	0.041	ns	ns
University education (EDU)	ns	0.005	0.037
Smoking (SMK)	0.001	<0.001	<0.001
Body Mass Index (BMI)	0.005	0.002	0.023
Alcohol consumption (ALC)	<0.001	<0.001	<0.001
Number of comorbidities (COM)	<0.001	<0.001	<0.001

ns: not significant. ^§^ Results are derived from analysis of covariance; ns: not significant; ANX: anxiety as measured by General Well-being Schedule; DEP: depression as measured by General Well-being Schedule; PCS: Pain Catastrophizing Scale.

**Table 3 jcm-09-02073-t003:** Path model’s parameters (1).

				95% C.I.						
		Estimate	Standardized Regression Coefficients B_std_. (2)	Lower	Upper	Std. Err	*z*-Value	*p*-Value	Std.lv (3)	R^2^
PCS ~										0.309
	WOM	0.305	0.021	−0.222	0.832	0.269	1.134	0.257	0.305	
	EDU	−0.493	−0.035	−1.001	0.015	0.259	−1.903	0.057	−0.493	
	CSM	1.025	0.040	0.007	2.044	0.520	1.973	0.049	1.025	
	EXS	0.599	0.041	0.060	1.137	0.275	2.180	0.029	0.599	
	OVW	0.080	0.005	−0.461	0.621	0.276	0.289	0.773	0.080	
	OBI	0.090	0.004	−0.708	0.888	0.407	0.220	0.826	0.090	
	OBS	2.002	0.051	0.437	3.567	0.799	2.507	0.012	2.002	
	LAC	−1.134	−0.077	−1.754	−0.515	0.316	−3.589	0.000	−1.134	
	HAC	−0.951	−0.032	−2.095	0.193	0.584	−1.629	0.103	−0.951	
	COM	0.238	0.049	0.049	0.427	0.097	2.463	0.014	0.238	
	PNI	0.857	0.234	0.706	1.009	0.077	11.092	0.000	0.857	
	ISI	0.107	0.052	0.023	0.190	0.043	2.511	0.012	0.107	
	ANX	0.491	0.324	0.427	0.555	0.033	15.014	0.000	0.491	
	DEP	0.246	0.125	0.166	0.326	0.041	6.013	0.000	0.246	
ANX~										0.163
	AGE	0.046	0.073	0.018	0.074	0.014	3.199	0.001	0.046	
	WOM	0.592	0.062	0.209	0.975	0.196	3.028	0.002	0.592	
	MRD	0.113	0.012	−0.268	0.493	0.194	0.580	0.562	0.113	
	CSM	0.643	0.038	−0.114	1.401	0.387	1.664	0.096	0.643	
	EXS	0.060	0.006	−0.329	0.449	0.198	0.303	0.762	0.060	
	OVW	−0.458	−0.047	−0.851	−0.065	0.201	−2.283	0.022	−0.458	
	OBI	−0.323	−0.025	−0.889	0.242	0.288	−1.121	0.262	−0.323	
	OBS	0.124	0.005	−0.907	1.156	0.526	0.236	0.813	0.124	
	LAC	−0.773	−0.079	−1.198	−0.348	0.217	−3.562	0.000	−0.773	
	HAC	0.826	0.042	−0.092	1.744	0.468	1.763	0.078	0.826	
	COM	0.390	0.122	0.252	0.527	0.070	5.558	0.000	0.390	
	PNI	0.397	0.165	0.293	0.502	0.053	7.462	0.000	0.397	
	ISI	0.313	0.230	0.252	0.373	0.031	10.174	0.000	0.313	
DEP~										0.171
	AGE	0.082	0.169	0.061	0.103	0.011	7.751	0.000	0.082	
	WOM	0.298	0.040	0.003	0.594	0.151	1.977	0.048	0.298	
	EDU	−0.392	−0.054	−0.677	−0.108	0.145	−2.704	0.007	−0.392	
	CSM	0.821	0.063	0.266	1.376	0.283	2.901	0.004	0.821	
	EXS	0.226	0.030	−0.072	0.523	0.152	1.486	0.137	0.226	
	OVW	−0.356	−0.048	−0.660	−0.052	0.155	−2.297	0.022	−0.356	
	OBI	−0.091	−0.009	−0.525	0.344	0.222	−0.408	0.683	−0.091	
	OBS	0.448	0.022	−0.407	1.304	0.436	1.027	0.304	0.448	
	LAC	−0.759	−0.101	−1.097	−0.422	0.172	−4.405	0.000	−0.759	
	HAC	0.630	0.042	−0.087	1.346	0.366	1.722	0.085	0.630	
	COM	0.267	0.108	0.164	0.370	0.052	5.096	0.000	0.267	
	PNI	0.172	0.092	0.094	0.250	0.040	4.324	0.000	0.172	
	ISI	0.227	0.216	0.182	0.273	0.023	9.852	0.000	0.227	

(1) TLI = 0.992, NFI = 0.998, NNFI = 0.992, CFI = 0.999, GFI = 0.998, SRMR = 0.005, RMSEA = 0.016. CI: Confidence Interval; PCS: Pain Catastrophizing Scale; ANX: anxiety as measured by General Well-being Schedule; DEP: depression as measured by General Well-being Schedule; PNI: Pain Intensity as measured by numeric rating scale for the previous seven days; ISI: Insomnia Severity Index; WOM: Women; MRD: Married; EDU: University education; CSM: Current smokers; EXS: Ex-smokers; OVW: Overweight; OBI: Obesity I; OBS: Obesity II and III; LAC: Low alcohol consumption; HAC: High alcohol consumption; COM: Number of comorbidities. (2) Completely standardized solution (estimates of parameters if the variances are unity). (3) Dependent Variable is standardized.

**Table 4 jcm-09-02073-t004:** Results of path analysis (direct effects)-schematic presentation.

Factor/Scale	ANX	DEP	PCS
Age (AGE)	↑	↑	ns
Women (WOM)	↑	↑	ns
Married (MRD)	ns	ns	ns
University education (EDU)	ns	↓	ns
Current smokers (CSM)	ns	↑	↑
Ex-smokers (EXS)	ns	ns	↑
Overweight (OVW)	↓	↓	ns
Obesity I (OBI)	ns	ns	ns
Obesity II and III (OBS)	ns	ns	↑
Low alcohol consumption (LAC)	↓	↓	↓
High alcohol consumption (HAC)	ns	ns	ns
Number of comorbidities (COM)	↑	↑	↑
PNI	↑	↑	↑
ISI	↑	↑	↑
ANX			↑
DEP			↑

ns: not significant; ↑: positive relationship; ↓: negative relationship. PCS: Pain Catastrophizing Scale; PNI: Pain Intensity as measured by numeric rating scale for the previous seven days; ANX: anxiety as measured by General Well-being Schedule; DEP: depression as measured by General Well-being Schedule; ISI: Insomnia Severity Index.

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
