# Peer review of "Pain Catastrophizing in Older Adults with Chronic Pain: The Mediator Effect of Mood Using a Path Analysis Approach"

_jcm, 2020, doi:10.3390/jcm9072073_

Round 1
Reviewer 1 Report
Carefully addressing pain catastrophizing in elderly is to be applauded.
Authors may be more careful in attributing effects and causation in the Results and Discussion. Well stated among the limitations but may consider revise some of the statements (i.e. attributing alcohol use a mechanism to cope with pain or pain catastrophizing)
Author Response
- Response: Thank you for your advice. We have modified the description in the revised manuscript. We also added our concern about uncertain development of alcohol consumption and pain, mood, and pain catastrophizing in the limitation. However, we would like to mention a recently meta-analysis which provides robust evidence for the analgesic properties of alcohol, which could potentially contribute to alcohol misuse in pain patients. Strongest analgesia occurs for alcohol levels exceeding World Health Organization guidelines for low risk drinking and suggests raising awareness of alternative, less harmful pain interventions to vulnerable patients may be beneficial
Thompson, T.; Oram, C.; Correll, C. U.; Tsermentseli, S.; Stubbs, B., Analgesic Effects of Alcohol: A Systematic Review and Meta-Analysis of Controlled Experimental Studies in Healthy Participants. The journal of pain: official journal of the American Pain Society 2017, 18 (5), 499-510.

Reviewer 2 Report
Figure 2 should be eliminated: is it too confused, and the same results are already clearly illustrated in table 3.
Author Response
- Response: Thank you for your suggestion. Point taken. We have deleted the figure and relevant notes in the text.

Reviewer 3 Report
The manuscript addresses the mediator effect of mood on the relationship between pain and catastrophizing in older people suffering chronic pain. Knowing the processes that maintain dysfunction and reduce the quality of life in these people is a relevant and unresolved issue.
As part of a population study, the main strength is the magnitude and representativeness of the sample. The methodology used is appropriate to achieve the aims. An explanatory- transversal-associative strategy with observed variables corresponds, in effect, to a mediation model.
My main objection to the manuscript is theoretical. In the introduction, the authors describe various empirical studies on the association between catastrophizing and emotions or mood. However, it is essential to frame the work in the fear-avoidance model. Two references that could include are:
Vlaeyen, J. W. S., & Linton, S. J. (2000). Fear-avoidance and its consequences in chronic musculoskeletal pain: A state of the art. Pain, 85, 317–332.
Vlaeyen JWS, Crombez G. Behavioral Conceptualization and Treatment of Chronic Pain. Annu Rev Clin Psychol 2020;16:187–212.
Likewise, on page 2, lines 56-59, the authors acknowledge that there is more evidence of catastrophizing as a mediator variable between pain and mood, and on the contrary, they only include a reference for the approach they defend (mood is the mediator variable). Considering that cognitive models arose precisely to highlight the mediator role of cognitive processes between situations and emotions-behaviours, the authors should better justify the mediation models they propose.
Minor Comments:
- An error is reported in the response alternatives of the Pain Catastrophizing Scale (PCS) that the subjects answered. The authors should clarify whether this fact could lead to any bias in the study results.
- The reliability of the GWBS and the ISI measures is not reported.
- The response rate to the survey was 66% (page 5, line 190). What implications could this fact have on the results?
- Page 7, line 212: missing legends of the variables in Figure 1.
- In Table 4 (page 10), being overweight is associated with less anxiety and depression. How is this result explained?
Author Response
The manuscript addresses the mediator effect of mood on the relationship between pain and catastrophizing in older people suffering chronic pain. Knowing the processes that maintain dysfunction and reduce the quality of life in these people is a relevant and unresolved issue. As part of a population study, the main strength is the magnitude and representativeness of the sample. The methodology used is appropriate to achieve the aims. An explanatory- transversal-associative strategy with observed variables corresponds, in effect, to a mediation model.
My main objection to the manuscript is theoretical. In the introduction, the authors describe various empirical studies on the association between catastrophizing and emotions or mood. However, it is essential to frame the work in the fear-avoidance model. Two references that could include are:
Vlaeyen, J. W. S., & Linton, S. J. (2000). Fear-avoidance and its consequences in chronic musculoskeletal pain: A state of the art. Pain, 85, 317–332.
Vlaeyen JWS, Crombez G. Behavioral Conceptualization and Treatment of Chronic Pain. Annu Rev Clin Psychol 2020;16:187–212.
Likewise, on page 2, lines 56-59, the authors acknowledge that there is more evidence of catastrophizing as a mediator variable between pain and mood, and on the contrary, they only include a reference for the approach they defend (mood is the mediator variable). Considering that cognitive models arose precisely to highlight the mediator role of cognitive processes between situations and emotions-behaviours, the authors should better justify the mediation models they propose.
- Response: We agree with the reviewer. We have now added that “Based on the fear avoidance model cognitive-behavioral risk factors play a crucial role in the development and persistence of pain. However, there is ongoing evidence supporting the necessity for modifications on the model to address how multi-dimensional factors may interact” Page 5
Wideman TH, Adams H, Sullivan MJ. A prospective sequential analysis of the fear-avoidance model of pain. Pain. 2009;145:45–51.
Wideman TH, Asmundson GGJ, Smeets R J, Zautra AJ, Simmonds MJ, Sullivan MJL, Haythornthwaite JA, Edwards RR. Re-Thinking the Fear Avoidance Model: Toward a Multi-Dimensional Framework of Pain-Related Disability. Pain. 2013 Nov; 154(11):2262-5.
Minor Comments:
- An error is reported in the response alternatives of the Pain Catastrophizing Scale (PCS) that the subjects answered. The authors should clarify whether this fact could lead to any bias in the study results.
- Response: Thank you for your suggestion. Point taken. We have now added in discussion that “Finally, the technical error in the response alternatives of the Pain Catastrophizing Scale accompanied by the medium response rate may underestimate our results” Page 16.
- The reliability of the GWBS and the ISI measures is not reported.
- Response: Point taken. We have added the references for the two instruments:
Bastien, C. H., Vallieres, A., & Morin, C. M. (2001). Validation of the Insomnia Severity Index as an outcome measure for insomnia research. Sleep Medicine, 2, 297– 307.
Morin, C. M., Belleville, G., Belanger, L., & Ivers, H. (2011). The Insomnia Severity Index: Psychometric indicators to detect insomnia cases and evaluate treatment response. Sleep, 34, 601– 608
Wang, M., Wang, S., Zhang, X., Xia, Q., Cai, G., Yang, X., … Pan, F. (2016). Analysis on the situation of subjective well‐being and its influencing factors in patients with ankylosing spondylitis. Health and Quality of Life Outcomes, 14, 118.
Leonardson, G. R., Daniels, M. C., Ness, F. K., Kemper, E., Mihura, J. L., Koplin, B. A., & Foreyt, J. P. (2003). Validity and reliability of the general well‐being schedule with northern plains American Indians diagnosed with type 2 diabetes mellitus. Psychological Reports, 93, 49– 58.
- The response rate to the survey was 66% (page 5, line 190). What implications could this fact have on the results?
- Response: Thank you for your concern. As usual, response rate in epidemiology studies of older people is higher than other age populations. However, we must consider non-response bias since frail older people often have severe impaired cognitive function and the questionnaire with focus of pain and consequences could be very difficult for this group. Moreover, the study procedure might have biased the voluntary sample, because the study design unintentionally selected participants who were strongly interested in the pain and pain-related issues. We have added this aspect in the discussion/limitation.
- Page 7, line 212: missing legends of the variables in Figure 1.
- Response: Point taken. We have added the legends of all the variables.
- In Table 4 (page 10), being overweight is associated with less anxiety and depression. How is this result explained?
- Response: thank you for your advice. Literature about BMI and mood is not consistent. In older adults, being underweight, overweight, or obese increased the risk of having clinically relevant depressive symptoms (Kim, J et al 2014, Lupino, FS et al 2010, Roberts, RE et al 2003). However, there were also studies that showed BMI were not associated with mental health (Bruin, MC et al 2018, Giuli, C et al, 2014, Mannucci, E 2010). A possible reason is the extensive adjustment of confounders in the various analysis. Another speculation is the intercorrelated relationships with other selected factors in the analysis. In our study, we suggested weight control based on the results that negative effect of obesity as well as the available evidence to date about weight management for the older adults (Porter Starr, KN et al 2016).
Kim J, Noh JW, Park J, Kwon YD. Body mass index and depressive symptoms in older adults: a cross‐lagged panel analysis. PLoS One. 2014; 9(12):e114891.
Luppino FS, de Wit LM, Bouvy PF, et al. Overweight, obesity, and depression. A systematic review and meta‐analysis of longitudinal studies. Arch Gen Psychiatry. 2010; 67(3): 220‐ 229.
Roberts RE, Deleger S, Strawbridge WJ, Kaplan GA. Prospective association between obesity and depression: evidence from the Alameda County Study. Int J Obes Relat Metab Disord. 2003; 27(4): 514‐ 521.
Bruin, MC, Comijs, HC, Kok, RM, Van der Mast, RC, Van den Berg, JF. Lifestyle factors and the course of depression in older adults: A NESDO study. Int J Geriatr Psychiatry. 2018; 33: 1000– 1008. https://doi.org/10.1002/gps.4889
Giuli, C., Papa, R., Bevilacqua, R. et al. Correlates of perceived health related quality of life in obese, overweight and normal weight older adults: an observational study. BMC Public Health 14, 35 (2014). https://doi.org/10.1186/1471-2458-14-35
Mannucci E, Petroni ML, Villanova N, Rotella CM, Apolone G, Marchesini G, QUOVADIS Study Group: Clinical and psychological correlates of health-related quality of life in obese patients. Health Qual Life Outcomes. 2010, 8: 90-10.1186/1477-7525-8-90.
Porter Starr, K. N., McDonald, S. R., Weidner, J. A., & Bales, C. W. (2016). Challenges in the Management of Geriatric Obesity in High Risk Populations. Nutrients, 8(5), 262. https://doi.org/10.3390/nu8050262
